## [Transparent Peer Review file · Nature Communications]

Excess FGFR3 signaling in achondroplasia disrupts turnover of resting zone chondrocytes via CREB signaling

Corresponding Author: Professor Noriyuki Tsumaki

Version 0:

Reviewer comments:

Reviewer #1

(Remarks to the Author)

This is an interesting study with high potential clinical relevance, providing novel clinical and molecular insights into the pathogenesis of achondroplasia. Nanao et al. were the first to demonstrate that *Fgfr3* mutations result in the expansion and reduced proliferation of resting zone chondrocytes. These mutations also cause aberrant progression of resting chondrocytes to columnar chondrocytes. Nanao et al. further provided single-cell RNA sequencing results on *Fgfr3*Ach mutant chondrocytes, revealing abnormal *Creb* signaling in the resting zone chondrocytes. Finally, they suggested that administration of the inhibitor 666-15 can partially alleviate the pathogenesis of *Fgfr3*Ach achondroplasia.

Despite the study's high interest and potential, the manuscript in its current form lacks robustness and has several limitations that weaken the conclusiveness of its findings.

Major concerns:

1. The *Fgfr3*-Achondroplasia model was first developed in 1999 and has been widely used since. It is surprising that no one has previously reported the resting zone phenotype. In some published histological images of the mutant growth plate, there is no expansion of the resting zone (e.g., <https://doi.org/10.1038/s41598-017-18801-0>). How do the authors explain this discrepancy? Have other papers reported an expanded resting zone? Is this phenotype specific to the novel mouse strain created in this study? If so, what are the differences? Could there be a risk of off-target mutations in the new strain?
2. The lethality in this novel mouse strain is higher than previously reported. Can this be explained? How does it align with previous publications? Do the mice exhibit tooth abnormalities? Are they kept on a soft diet?
3. The development of the secondary ossification center is delayed in *Fgfr3*-Ach mice. The authors should demonstrate that the expanded resting zone is not simply residual epiphyseal cartilage that has not yet been used for the formation of the secondary ossification center at the studied developmental window (P18–P21). This can be addressed by providing a more detailed characterization of SOC development, including additional markers for resting and hypertrophic chondrocytes, more precise imaging, and studying more time points around this developmental window.
4. The mechanism behind the expanded resting zone is unclear. The authors propose that limited recruitment of chondroprogenitors into the columnar zone may cause cellular accumulation in the resting zone. However, decreased proliferation in the resting zone should theoretically mitigate this effect. Reduced proliferation is currently judged only by EdU retention experiments (Fig. 2b). It is unclear if more cells were labeled immediately after multiple EdU injections or what the immediate proliferative rate in the resting zone at P21 is (Fig. 2a provides quantitative data only for the columnar zone, not the resting zone). Short-term tracing with Confetti mice could be an additional, independent, and powerful approach to assessing proliferation rates in various zones.
5. In connection with the previous point, does the expression of *Ihh* and PTHrP change in *Fgfr3*-Ach mice? These morphogens control the recruitment of resting zone cells into the columnar layer and could lead to the accumulation of cells in the resting zone.

6. The growth plate shrinks by P28, including the resting zone. What happens to the resting zone cells at that point? Do they undergo apoptosis? Do they differentiate directly into hypertrophic cells without a transit-amplifying stage?
7. Another major concern is the direct effect of FGFR3 on the resting zone. Do these cells express the FGFR3 receptor? Can this be demonstrated immunohistologically or through scRNA-seq data? If it is not a direct effect, can *Ihh* be involved?
8. The rescue experiments using the CREB inhibitor are valuable, but the rescue effect is only partial. The clinical relevance of the study could be enhanced if the authors combined the CREB inhibitor with vosoritide. Simultaneously targeting both the PKA/CREB and MAPK pathways may provide a more effective therapeutic strategy.
9. Many of the experiments and models are poorly characterized throughout the study. For example:
 - o Figure 1: See my comments in #3 and #4 above.
 - o Figure 2: Other zones are not quantified, the definition of zones is lacking, and there is no data on the transition of labeled cells between zones, etc.
 - o Figure 3: Poor image quality and quantification; initial labeling is missing, etc.
 - o Figure 4 and supplementary figures: scRNA-seq data are superficially analyzed and presented.
 - o Figures 1–4: Additional markers for the resting zone, such as APOE, SFRP5, and PTHrP, are missing.
 - o Figure 6: Various zones are not quantified, and markers of hypertrophy, stem cells, and proliferation are all lacking.

Minor comments to strengthen the study:

1. One piece of information missing from the 666-15 CREB signaling inhibition experiment is whether 666-15 administration reduces p-CREB signaling in *Fgfr3*^{Ach} mice treated with 666-15. In Figure 6, only Spondin1 and CD73 staining are shown.
2. If increased CREB activity in *Fgfr3*^{Ach} mice resting chondrocytes is true, SCENIC analysis of the scRNA data should reveal a consistent increase in CREB transcription factor activity. The authors could provide this data as further support.
3. The justification of cluster 3 as resting chondrocytes corresponding to the expanded resting zone is relatively weak. Although cluster 3 likely represents resting chondrocytes, its proximity to Col10a1+ cluster 2, which also increases in number in *Fgfr3*^{Ach} mice, undermines the reliability of the finding. Additionally, the Spondin1 staining in Figure 4 is not entirely convincing. Some "hypertrophic chondrocytes" can be seen in the *Fgfr3*^{Ach} mutants close to the SOC. Staining with one more marker for cluster 3 and another for cluster 2 would strengthen this finding.
4. The term "migration toward the primary ossification center" is debatable. A more conservative term, such as "reduced turnover of chondrocytes," may be more accurate.
5. Many of the histology images in the main figures are of low quality, making it difficult for reviewers to assess the authenticity of the data.
6. Quantification of several immunohistological images will be helpful, like Figs 4c, 5b, 6e.
7. Currently, figures are poorly presented and cannot be read without consulting figure legends and main text. Information like the time point of analysis, duration of treatment, and time of tamoxifen injections can all be incorporated into the figures for easy navigation.

Reviewer #2

(Remarks to the Author)

Reviewer #3

(Remarks to the Author)

This is an interesting paper that presents a new model for the mechanism behind the achondroplasia mutation in *Fgfr3*. Using ScRNAseq authors have identified the up-regulation of the CREB pathway as an effector of the achondroplasia phenotype. The rescue of the mouse model through the administration of 666-15, a CREB inhibitor, further supports this observation. These observations support further investigation of the CREB pathway as a therapeutic target for treating *Fgfr3*-Ach induced achondroplasia.

There are some concerns that the authors need to address.

1. It is not clear why the first time point analyzed was P18-P21. At what age does one first see the *Fgfr3*-Ach phenotype, especially the increase in expansion of the resting zone chondrocytes and how much time since then does one see a size/bone phenotype. A temporal analysis that includes late embryonic and neonatal time points is strongly suggested.
2. Is the expansion in resting zone chondrocytes seen also in other mouse models that have investigated *Fgfr3*-Ach or is it unique to this mouse created by this lab? If it is unique then it would suggest that the genetic background of the mice is also contributing to this phenotype and not just the p.Gly374Arg mutation. In this mouse model, the description provided by the authors suggest that the genetic background is essentially C57BL/6. Is this different from other Ach models created by other labs. The authors do not discuss why this is not seen in other labs as it is a very distinguishing feature of their mouse.
3. The effect on the resting zone is potentially very interesting; however, the resting zone is poorly defined. Additional markers for resting zone chondrocytes should be used to determine if they are really looking at resting zone expansion and where they are drawing their boundaries. In situ hybridization or immunostaining for PthrP or Gas1 (see Hallet et al., eLife 2021 10.7554/eLife.64513) in control and Ach mice should be used to confirm expansion of the resting zone. How does the

expression of PthrP and Gas1 compare to that of Spondin1 in Fgfr3-Ach mice?

4. Resting zone chondrocytes may have some stem cell properties, but they are chondrocytes and not stem cells and the authors need to keep this mind when writing about this zone and not equate the two. The cells in this zone are chondrocyte progenitors at best but not stem cells. Also, a stem cell usually is not defined by the presence of one marker but a combination of markers.
5. From the ScRNAseq clusters, they essentially ignore Cluster 2 even though, like Cluster 3, it is also enriched in mutant cells. From Sup Fig 3b one can see that Cluster 2 shows the biggest change in numbers in all 3 mice that were used for this analysis and not cluster 3. Some analysis and discussion of cluster 2 should be provided.
6. The authors need to follow the increase of Fgfr3 activation or its decrease in the rescued mice by looking at additional downstream targets of Fgfr3 such as Stat1 or Snail.
7. With few exceptions, the IHC experiments are not very visually convincing. The choice of colorimetric IHC shows very weak expression and it is very hard to see the positive signals because the contrast of the positive signals with the secondary stain is not very high. In several cases it is very difficult to see a signal above background (CBP, p-FRS2). Spondin1 IHC in Fig. 6 is not convincing.
8. The model says that resting zone chondrocytes are unable to proliferate. The experiments with EDU probably influenced this conclusion. But this is not consistent with expansion of the resting zone. Some discussion of this point and maybe refinement of the model would help. Where are these cells coming from? Can they be identified in a lineage trace? This is the reason why a temporal experiment showing how the morphology and expression of resting zone markers changes with time is important to understand how resting zone expansion is taking place.
9. The DISCUSSION Section is very weak. It is necessary to discuss these results in the context of other models for Fgfr3-Ach and how the current model is similar and/or different.
10. Figure 6a. Control WT Data is missing. Need to know if the rescue is partial or complete. Figure 6b suggests a partial rescue.
11. Control data is missing in Fig 6 for administration of 666-15 to WT mice. If 666-15 is administered to WT mice (as shown for Fgfr3-Ach in Fig 6F), do control WT mice show more expression for CD73?
12. Sentence 816: Figure 6 legend should say "666-15 partially rescued the Phenotype" as the bones showed elongation but did not reach WT levels in Fig 6b.
13. Fig 6e. The immunostaining for Spondin1 is essentially not visible to the naked eye. The authors need a better figure. I suggest that they try immunofluorescence and confocal imaging for all their immunohistochemistry analysis.
14. It would have been interesting to see if any cells in the expanded resting zone in Fgfr3-Ach mice are dying by apoptosis. Did the authors perform any TUNEL analysis of the growth plate?

Minor Issues:

1. Sentence 48-49: "within the growth plate cartilage". This reads like a compound Statement. Also, statement is scientifically wrong.
2. Sentence 53, 151, 194: As pointed out above, resting zone chondrocytes are not stem cells.
3. Sentence 163: "by mating Fgfr3 Ach mice: The authors did not mate Ach mice as these mice die by 6 weeks. They introduced the Col11a3-CreERT in the parents' mating profile. Sentence needs to be corrected.
4. 176: The authors need to add here that the schema is shown in Fig 3C.
5. 188: how do the authors know that it is a partial closure? Is this conjecture or did they investigate this?
6. 198: not "resting" but "resting zone chondrocyte abnormalities".
7. 208: perhaps more appropriate to say "hematopoietic markers" as opposed to blood cell markers.
8. 219: not just Cluster 0 but cluster 1 also has strong expression of markers Sp7 and Pth1r as is evident from Figure. So cluster 1 is unlikely to be proliferative chondrocytes.
9. 226: "enriched in" rather than "enriched with".
10. 255: hypertrophic zone: not shown in Figure
11. 272: Typo: not "maker" but "marker"
12. 278: not FGFR.
13. 470: The cells were Lysed, plated....?? Why were the cells lysed. If cells were lysed, what did the authors grow/plate?
14. 727: 5 mm in (a)? or (b)?
15. 731: Stem cell chondrocytes. See above. Stem cells are not chondrocytes. This is a misnomer.
16. 779: The authors need to write what color is Spondin1 after the IHC? Brown? Blue? What should we look for?
17. Figure 5g: there is a qPCR analysis shown. But its description is missing in legends (Sentence 806-809)
18. Supplementary Fig. 3. 875: How was the ratio calculated? What was the denominator?

Reviewer #4

(Remarks to the Author)

Version 1:

Reviewer comments:

Reviewer #1

(Remarks to the Author)

Dear Editor,

The authors have addressed all of my concerns in a clear, thorough, and detailed manner. In my view, the manuscript has been substantially improved. It now reads more smoothly; the data are presented more comprehensively; the figures are of higher quality; and the additional analyses enable more robust conclusions. The clarified figure presentation further enhances readability. My congratulations to Tsumaki-sensei and his team for this rigorous revision, the exciting narrative, and the important discoveries it contains. I hope the authors will continue this line of research and explore combinatorial therapeutic approaches.

I now find the evidence very convincing that achondroplasia is associated with impaired behavior of progenitor cells in the resting zone. This is a highly novel and important finding, shedding new light on the mechanisms by which FGFR3 activation leads to growth retardation. The mechanisms uncovered here may ultimately support the development of new therapeutic approaches for children with achondroplasia.

Reviewer #2

(Remarks to the Author)

Reviewer #3

(Remarks to the Author)

Overall, the manuscript is significantly improved and the revised IHC data is superior in quality and allows for improved confidence in the observation that it is the resting zone in Fgfr3Ach mouse growth plate that is enlarged, a major conclusion from this paper. This observation was also helped by the ScRNAseq data set that identified Spondin1 as a unique marker of the resting phase. The improved IHC allowed for better visibility of the enlarged resting phase and also increased confidence in this conclusion. Likewise, the new Stat5ab IHC shown in this manuscript also supports this observation.

The other major conclusion is the identification of the CREB pathway as being activated in the resting zone and its contribution to preventing chondrocytes from differentiation into proliferative chondrocytes. However, the authors are also modeling that the CREB pathway is inhibiting the self-renewal properties of the resting zone chondrocytes. But if the CREB pathway is inhibiting self-renewal then EdU incorporation into the resting zone as seen in Figure 2 would not have increased.

The long term EdU labeling in Figure 2 Control also does not make sense. Where do the EdU labeled proliferating chondrocytes go. At least some should be seen in the hypertrophic zone or in trabecular osteoblasts that are derived from hypertrophic chondrocytes.

From analysis of ScRNAseq data, the authors also conclude that cluster #3 (resting zone cluster) gives rise to cluster #2, which means that the cluster #2 zone is not contributing to the resting zone. The resting zone is enlarged, and these cells must come from somewhere. The increased EdU in this zone suggests self-renewal and failure to differentiate into proliferating chondrocytes, which is at odds with the CREB model in figure 7 of decreased self-renewal. It seems that the model in Figure 7 is not consistent with the model in Figure 3c, suggesting that the Figure 7 model is not correct. The only conclusion that seems supported by the data is that the CREB Pathway is preventing the resting zone chondrocytes to differentiate into proliferative chondrocytes which results in enlarging the resting zone.

Even though the authors claim in their rebuttal that loss of self-renewal is only a model in Fig 3c, the abstract (line 34) claims this as a conclusion. This needs to be modified.

The authors have also included embryonic data and data of neonatal mice that show that the enlargement of the resting zone does not begin until after P18. A significant observation as the transition from normal resting zone to enlarged resting zone is happening much later in neonatal mice. But there is no explanation for this phenomenon or even discussion of this observation in the Discussion section. Other mouse models for achondroplasia show changes in growth plate histology at earlier ages. What happens to the CREB pathway in neonatal mice before P18 when expansion of resting zone is not seen? Is the CREB pathway normal. This should be tested by IHC as the authors have done in Fig. 5a. Negative data from this analysis would further validate the involvement of the CREB pathway in this observation.

Minor:

1. Fig. 1d, should be Survival not Probability of survival

2. Fig 2 legend:

Line 919. The authors use Click-iT chemistry to identify EdU positive cells. But writing "immunostained with anti-EdU antibody" here is wrong.

3. Fig. 3b show where insets come from. 3d, define SOFG

4. Line 270 and 1148 and also elsewhere: "violin plots" and not "viline"

5. Line 284 Omd is Osteomodulin, not Ostomodulin

6. Legend missing for Supplementary Fig. 9c

Reviewer #4

(Remarks to the Author)

Version 2:

Reviewer comments:

Reviewer #3

(Remarks to the Author)

The authors have addressed my previous comments. No further comments.

Reviewer #4

(Remarks to the Author)

Detailed Point-by-Point Response to Reviewers' Comments

Dear reviewers,

We would like to thank all of the reviewers for their time and helpful comments. Following those comments, we have made changes to the manuscript (presented in red font). Below are our point-by-point responses (in black font) to the reviewers' comments (in blue font).

Reviewer #1 (Remarks to the Author): This is an interesting study with high potential clinical relevance, providing novel clinical and molecular insights into the pathogenesis of achondroplasia. Nanao et al. were the first to demonstrate that Fgfr3 mutations result in the expansion and reduced proliferation of resting zone chondrocytes. These mutations also cause aberrant progression of resting chondrocytes to columnar chondrocytes. Nanao et al. further provided single-cell RNA sequencing results on Fgfr3Ach mutant chondrocytes, revealing abnormal Creb signaling in the resting zone chondrocytes. Finally, they suggested that administration of the inhibitor 666-15 can partially alleviate the pathogenesis of Fgfr3Ach achondroplasia.

Despite the study's high interest and potential, the manuscript in its current form lacks robustness and has several limitations that weaken the conclusiveness of its findings.

Major concerns:

1. The Fgfr3-Achondroplasia model was first developed in 1999 and has been widely used since. It is surprising that no one has previously reported the resting zone phenotype. In some published histological images of the mutant growth plate, there is no expansion of the resting zone (e.g., <https://doi.org/10.1038/s41598-017-18801-0>). How do the authors explain this discrepancy? Have other papers reported an expanded resting zone? Is this phenotype specific to the novel mouse strain created in this study? If so, what are the differences? Could there be a risk of off-target mutations in the new strain?

<Authors' response>

We agree that resting zone phenotypes were not described in the text in previous reports. Therefore, we looked into the Figures of previously reported papers.

In the study by Shazeeb et al.) [1], Figure 6G indicates the histology of 3-week-old proximal tibia growth plates. Proliferative (red bracket) and hypertrophic (yellow

bracket) zones in the Ach mouse compared to the WT mouse, respectively, are indicated. As seen in Figure 6G, the length of the resting zone above the red brackets appears slightly longer in the Ach mouse compared to WT mouse. The Ach mouse model used in this

[FIGURE REDACTED]

study is the transgenic model expressing mouse *Fgfr3* containing the achondroplasia mutation under the *Col2* promoter[2]. The phenotype of this transgenic mouse model is milder than that of human achondroplasia, probably because the *Col2* promoter did not recapitulate expression patterns of *Fgfr3* gene. This would explain why the degree of expansion of the resting zone is undetectable or mild in this transgenic mouse model. In addition, the difference between the FVB/NJ background in the transgenic mouse model and C57Bl/6 background in our study may underlie this reason.

Shazeeb et al. (<https://doi.org/10.1038/s41598-017-18801-0>), Figure 6G

Wang et al. 1999 (<https://doi.org/10.1073/pnas.96.8.4455>) reported conditional knock-in mice in which G374R (the ortholog of human G380R) was introduced into *Fgfr3* locus [3]. The mice in Wang et al.'s study are almost identical to ours except for that they used the Cre/loxP system while we used the Flp/frt system to make knock-in mice conditional. In the aforementioned study, Figures 5 c,d indicate the histology of 30-day-old proximal tibia growth plates. Although Wang et al. do not mention this in the text, the resting chondrocyte zone appears markedly expanded in the knock-in mice (Figure 5d, red bracket which we added to the original figure) compared to that of the wild-type

mouse (Figure 5c, red bracket we added to the original figure). The histology of 30-day-old knock-in mice in Wang et al.'s study is similar to that of our 21-day-old knock-in mice shown in Figure 1h in our manuscript. The difference between 129/SV background in Wang et al. knock-in mice and C57Bl/6 background in our mice may account for the difference in ages when the mice show typical phenotype.

Although the authors did not mention it in previous studies, their figures indicate that the previously reported Ach model mice also showed expansion of the resting chondrocyte zone similar to the histology shown by our mice. We have revised the Results section (lines 149-150) and added this information to the Discussion section (lines 385-388).

[FIGURE REDACTED]

Wang et al. (<https://doi.org/10.1073/pnas.96.8.4455>), Figures 5 c,d
Histological analysis of the growth plates from the tibia of neo-FGFR3 heterozygotes (FGFR3^{G374Rneo/+}) (d) and wild-type mice (c). We added resting zone (RZ) red brackets.

Our manuscript, Figure 1h; second row.

Histological images of growth plate cartilage in the proximal tibia of *Fgfr3^{Ach}* mice (right) and control mice (left) at P21. Dotted lines indicate the resting, proliferative, and

hypertrophic zones. SOC, secondary ossification center; RZ, resting zone; PZ, proliferative zone; HZ, hypertrophic zone; POC, primary ossification center.

2. The lethality in this novel mouse strain is higher than previously reported. Can this be explained? How does it align with previous publications? Do the mice exhibit tooth abnormalities? Are they kept on a soft diet?

<Authors' response>

Two mouse models express G374R (the mouse ortholog of human G380R). One mouse model is the transgenic mice mis-expressing *Fgfr3* cDNA with G374R mutation under the control of *Col2a1* promoter/enhancer sequences[2]. The phenotype of this transgenic mice appears milder than that of human patients with achondroplasia. This is probably because *Col2a1* promoter / enhancer sequences do not completely reproduce the expression pattern of *Fgfr3*.

The other mouse model is the conditional knock-in mice introduced by Wang et al[3]. Their mice are almost identical to ours except for that they used Cre/loxP system while we used Flp/frt system to make knock-in mice conditional. They state the weight of the mutant mice is less than 50% of that of the wild-type mice. Their mutant mice showed malalignment of the incisors. To allow for normal food intake, they shortened the teeth of the dwarfs, and softened the pelleted mouse chow by wetting. In addition, the growth plate shortening in their mutant mice was greater than that reported in most cases of human achondroplasia. Therefore, the phenotype of Wang et al.'s mutant mice is more severe than that of human patients with achondroplasia. The phenotype of our mice appears slightly more severe than that of Wang et al.'s mice because the histology of growth plate for 30-day-old mice corresponds to that of our 21-day-old mice as described above in the response to the comment #1. The background of Wang et al.'s mice is 129/SV while that of our mice is C57Bl/6. The difference in the backgrounds may be one of reasons for the slight difference in severities between the two mouse models. Our mutant mice also exhibited malalignment of incisors (the image below), and we shortened the teeth of the mice, and softened the pelleted mouse chow by wetting.

We have added this information to the Results (line 125-127; 131-132) and Methods sections (lines 456-458).

Malalignment of incisors of *Fgfr3^{Ach}* mice when teeth were not shortened.

3. The development of the secondary ossification center is delayed in *Fgfr3-Ach* mice. The authors should demonstrate that the expanded resting zone is not simply residual epiphyseal cartilage that has not yet been used for the formation of the secondary ossification center at the studied developmental window (P18–P21). This can be addressed by providing a more detailed characterization of SOC development, including additional markers for resting and hypertrophic chondrocytes, more precise imaging, and studying more time points around this developmental window.

<Authors' response>

We have added the histology of the proximal tibia of *Fgfr3^{Ach}* mice from 15.5 days postcoitum (E15.5) to P28 in Supplementary Figures 1 and 2. We have also performed immunohistochemistry for Col10, a marker for the boundary between the secondary ossification center and growth plate (Supplementary Figure 3). Formation of the secondary ossification center was confirmed via the expression of Col10 at the periphery of the secondary ossification center in *Fgfr3^{Ach}* mice at P18, indicating that the expanded resting zone is not simply residual epiphyseal cartilage. We have added this description to the Results section (lines 136-141; 154-156).

We also analyzed the expression of markers for resting zone chondrocytes such as *Clu1*, *ApoE*, and *Sfrp5* via scRNA-seq. These genes were preferentially expressed in Cluster #3. Immunohistochemical analysis for SFRP5 indicated expression in expanded resting zone chondrocytes in *Fgfr3^{Ach}* mice at P21. We have added this description and data to the Results section (lines 274-279) and Supplementary Fig. 10.

4. The mechanism behind the expanded resting zone is unclear. The authors propose that limited recruitment of chondro-progenitors into the columnar zone may cause cellular accumulation in the resting zone. However, decreased proliferation in the resting zone should theoretically mitigate this effect. Reduced proliferation is currently judged only by EdU retention experiments (Fig. 2b). It is unclear if more cells were labeled immediately after multiple EdU injections or what the immediate proliferative rate in the

resting zone at P21 is (Fig. 2a provides quantitative data only for the columnar zone, not the resting zone). Short-term tracing with Confetti mice could be an additional, independent, and powerful approach to assessing proliferation rates in various zones.

<Authors' response>

In Figure 2, we counted the numbers of EdU-positive cells in the resting, proliferative, and hypertrophic zones, and added the data in the graphs. The short-term EdU labeling assay indicated that the proliferation rates of resting zone chondrocytes did not differ significantly between *Fgfr3^{Ach}* mice and control mice. We have added this information to the Results section (lines 163-166) .

To confirm labeling in the long-term EdU labeling assay, we sacrificed mice at P8, one day after the consecutive labeling at P4-7. Chondrocytes were similarly labeled in the control and *Fgfr3^{Ach}* mice at P8. Based on these additional data, the expansion of the resting zone in *Fgfr3^{Ach}* mice is attributed to a decreased provision of proliferative zone chondrocytes from resting zone chondrocytes. Proliferation rates in resting zone chondrocytes in *Fgfr3^{Ach}* mice were not increased. We have added these data to the Results section (lines 176-181) and Supplementary Figure 4.

5. In connection with the previous point, does the expression of Ihh and PTHrP change in *Fgfr3-Ach* mice? These morphogens control the recruitment of resting zone cells into the columnar layer and could lead to the accumulation of cells in the resting zone.

<Authors' response>

scRNA-seq analysis indicated that there were neither obvious differences in the expression of Ihh in cluster #0 (prehypertrophic / hypertrophic zone chondrocytes) nor PTHrP in cluster #3 (resting zone chondrocytes) between the control and *Fgfr3^{Ach}* mice (Supplementary Fig. 12). However, we found that *Fgfr3* was highly expressed in the cluster #3 cells in *Fgfr3^{Ach}* mice compared to those in the control mice (Supplementary Fig. 12). Immunohistochemistry confirmed expression of FGFR3 in the expanded resting zone chondrocytes in *Fgfr3^{Ach}* mice (Fig. 5a). These results collectively suggest that the abnormalities of the *Fgfr3^{Ach}* resting zone is the direct effect of mutant FGFR3 signaling in these cells but not paracrine effects from Ihh-PTHrP morphogens. We have added this information to the Results section (lines 314-320) and to Figure 5a and Supplementary Figure 12.

6. The growth plate shrinks by P28, including the resting zone. What happens to the

resting zone cells at that point? Do they undergo apoptosis? Do they differentiate directly into hypertrophic cells without a transit-amplifying stage?

<Authors' response>

We performed TUNEL assay and did not observe increased staining in resting zone chondrocytes in *Fgfr3^{Ach}* mice compared to that in control mice. This result does not suggest that the depletion of *Fgfr3^{Ach}* resting zone chondrocytes at P28 was due to apoptosis. Reduced number of resting zone chondrocytes at P28 and loss of CD73 expression in *Fgfr3^{Ach}* mice suggests that the resting zone chondrocytes lost self-renew activity. In addition, the height of the proliferative zone was reduced in *Fgfr3^{Ach}* mice (Fig. 1h), suggesting that the resting zone chondrocytes in *Fgfr3^{Ach}* mice differentiate into hypertrophic zone chondrocytes through a reduced transit-amplifying stage. These changes would collectively lead to a decrease in the height of the entire growth plate cartilage. We have added data on the TUNEL assay to the Method section and Supplementary Figure 5, and added these results and speculation to the Results section (lines 211-214; 217-221).

7. Another major concern is the direct effect of FGFR3 on the resting zone. Do these cells express the FGFR3 receptor? Can this be demonstrated immunohistologically or through scRNA-seq data? If it is not a direct effect, can *Ihh* be involved?

<Authors' response>

As described in our response to comment #5, we performed additional experiments to analyze whether FGFR3 signaling is activated in expanded resting zone chondrocytes in *Fgfr3^{Ach}* mice. scRNA-seq analysis indicated that *Fgfr3* mRNA was highly expressed in cluster #3 cells in *Fgfr3^{Ach}* mice than in the control mice (Supplementary Fig. 12 a,b). Immunohistochemistry confirmed the expression of FGFR3 in the expanded resting zone chondrocytes in *Fgfr3^{Ach}* mice (Figure 5a). FRS2, a direct substrate of fibroblast growth factor receptors (FGFR), was phosphorylated in chondrocytes within the expanded resting zone of *Fgfr3^{Ach}* mice (Fig. 5a). These results collectively suggest the activation of FGFR3 signaling in these cells.

We have added this description to the Results section (lines 300-313) and to Figure 5a and Supplementary Figure 12.

8. The rescue experiments using the CREB inhibitor are valuable, but the rescue effect is only partial. The clinical relevance of the study could be enhanced if the authors combined

the CREB inhibitor with vosoritide. Simultaneously targeting both the PKA/CREB and MAPK pathways may provide a more effective therapeutic strategy.

<Authors' response>

Thank you for the valuable insight. Vosoritide targets proliferative zone chondrocytes and hypertrophic zone chondrocytes while 666-15 targets resting zone chondrocytes. Therefore, simultaneous administration of Vosoritide and 666-15 may have additive effects because they act on different cells and different pathways. We have added this information to the Discussion section (lines 4005-410). To analyze this effect in mouse model, we first need to determine optimal combination dosages of vosoritide and 666-15, which would take substantial time. We intend to perform this experiment in future research.

9. Many of the experiments and models are poorly characterized throughout the study. For example:

o Figure 1: See my comments in #3 and #4 above.

<Authors' response>

Please see our responses to comments #3 and #4 above.

o Figure 2: Other zones are not quantified, the definition of zones is lacking, and there is no data on the transition of labeled cells between zones, etc.

<Authors' response>

We additionally quantified numbers of labelled cells in the resting and hypertrophic zones respectively. We indicated data in these zones in a graph (Figure 2). We have added the definition of zones in the Materials section (lines 485-492).

o Figure 3: Poor image quality and quantification; initial labeling is missing, etc.

<Authors' response>

We substituted the initial images in Figure 3 with clearer ones. We added a scheme and labels that explain the timings of tamoxifen administration and sacrifice.

o Figure 4 and supplementary figures: scRNA-seq data are superficially analyzed and presented.

<Authors' response>

We performed additional analysis on scRNA-seq data including SCENIC analysis and RNA velocity analysis. Through SCENIC analysis, we detected *Creb1* regulons enriched in *Fgfr3^{Ach}* mice and cluster #3. RNA velocity analysis indicated that cluster #3 becomes cluster #2. We have added data to the Results section (lines 293-296; 325-327), Methods section (lines 561-580), and Supplementary Figures 11d and 15.

o Figures 1–4: Additional markers for the resting zone, such as APOE, SFRP5, and PTHrP, are missing.

<Authors' response>

We analyzed the expressions of *Clu*, *ApoE*, and *Sfrp5* using scRNA-seq data. Cluster #3 preferentially expressed these markers for resting zone chondrocytes.

We also performed immunohistochemical analysis for SFRP5. SFRP5 was specifically expressed in the expanded resting zone in *Fgfr3^{Ach}* mice and in the resting zone of control mice. These results collectively confirm the hypothesis that cluster #3 corresponds to cells in the expanded resting zone of chondrocytes in *Fgfr3^{Ach}* mice. We have added these results to the Results section (lines 274-279) and Supplementary Figure 10.

o Figure 6: Various zones are not quantified, and markers of hypertrophy, stem cells, and proliferation are all lacking.

<Authors' response>

We measured the thickness of each zones and added the data to Figures 2, 6e; and Supplementary Figure 1b. We performed additional immunohistochemical analysis for COL2 and COL10 and have added the data to the Results section (lines 361-367) and Figure 6d.

Minor comments to strengthen the study:

1. One piece of information missing from the 666-15 CREB signaling inhibition experiment is whether 666-15 administration reduces p-CREB signaling in *Fgfr3^{Ach}* mice treated with 666-15. In Figure 6, only Spondin1 and CD73 staining are shown.

<Authors' response>

We performed immunohistochemistry and found that the expression of phospho-CREB was reduced in the growth plate cartilage of *Fgfr3^{Ach}* mice treated with 666-15 compared to those treated with the vehicle. We have added these data to the Results section (lines 365-367) in Figure 6f.

2. If increased CREB activity in *Fgfr3^{Ach}* mice resting chondrocytes is true, SCENIC analysis of the scRNA data should reveal a consistent increase in CREB transcription factor activity. The authors could provide this data as further support.

<Authors' response>

We performed SCENIC analysis and detected *Creb1* regulons enriched in *Fgfr3^{Ach}* mice and cluster #3. We have added data to the Results section (lines 325-327), Methods section (lines 561-580), and Supplementary Figures 11d and 15.

3. The justification of cluster 3 as resting chondrocytes corresponding to the expanded resting zone is relatively weak. Although cluster 3 likely represents resting chondrocytes, its proximity to *Col10a1*+ cluster 2, which also increases in number in *Fgfr3^{Ach}* mice, undermines the reliability of the finding. Additionally, the *Spondin1* staining in Figure 4 is not entirely convincing. Some "hypertrophic chondrocytes" can be seen in the *Fgfr3^{Ach}* mutants close to the SOC. Staining with one more marker for cluster 3 and another for cluster 2 would strengthen this finding.

<Authors' response>

Thank you for this important point. In addition to cluster #3, cluster #2 was also enriched with *Fgfr3^{Ach}* mouse cells (Fig. 4a; Supplementary Fig. 8). To characterize cluster #2, we performed immunohistochemical analysis for markers for cluster #2 and RNA velocity assay. DEGs between cluster #2 and the remaining clusters included *Colla2* and *Omd* (Supplementary Fig. 11a). The *Omd* gene encodes OSTOMODULIN (also known as OSTEOADHERIN). The feature plot function confirmed that DEGs were preferentially expressed in cluster #2 (Supplementary Fig. 11b). Immunohistochemical analysis using an anti-COL1 or anti-OSTEOMODULIN antibodies indicated that they were specifically expressed in the boundary between the resting zone and secondary ossification center in both control and *Fgfr3^{Ach}* mice (Supplementary Fig. 11c). Cells in cluster #2 expressed *Coll10a1* (Supplementary Figs. 7c, 11b), and cells at the boundary between the resting zone and secondary ossification center were hypertrophic and expressed COL10 (Supplementary Fig. 3). These results suggest that cluster #2

corresponds to cells in the boundary between the resting zone and secondary ossification center. We selected clusters #2 and #3 cells and subjected them to trajectory inference and RNA velocity analysis. The result indicated that cluster #3 become cluster #2 (Supplementary Fig 9d). These results collectively suggest that primary lesion resides in cluster #3 rather than cluster #2. We have added these data to the Results section (lines 281-296) and Supplementary Fig. 11.

We performed immunohistochemical analysis for SPONDIN1 using fluorescence to obtain improved images. We have substituted one in Fig. 4d. We additionally performed immunohistochemical staining for SFRP5 (markers for cluster #3) and have added data to Supplementary Figure 10b.

4. The term “migration toward the primary ossification center” is debatable. A more conservative term, such as “reduced turnover of chondrocytes,” may be more accurate.

<Authors’ response>

We agree that the term “reduced turnover of chondrocytes” is more appropriate. We have revised this term in the abstract (line 35).

5. Many of the histology images in the main figures are of low quality, making it difficult for reviewers to assess the authenticity of the data.

<Authors’ response>

We substituted images of high quality (Figures 4d, 5a, and 6f). We employed fluorescence instead of DAB for detection in immunohistochemistry in several experiments. We also employed confocal microscopy.

6. Quantification of several immunohistological images will be helpful, like Figs 4c, 5b, 6e.

<Authors’ response>

We quantified immunohistological images in Figures 4d and 5b.

7. Currently, figures are poorly presented and cannot be read without consulting figure legends and main text. Information like the time point of analysis, duration of treatment, and time of tamoxifen injections can all be incorporated into the figures for easy navigation.

<Authors' response>

Thank you for the helpful suggestion. We have incorporated information such as the time point of analysis, duration of treatment, and time of tamoxifen injections into the Figures.

Reviewer #3 (Remarks to the Author): This is an interesting paper that presents a new model for the mechanism behind the achondroplasia mutation in *Fgfr3*. Using ScRNAseq authors have identified the up-regulation of the CREB pathway as an effector of the achondroplasia phenotype. The rescue of the mouse model through the administration of 666-15, a CREB inhibitor, further supports this observation. These observations support further investigation of the CREB pathway as a therapeutic target for treating *Fgfr3*-Ach induced achondroplasia.

There are some concerns that the authors need to address.

1. It is not clear why the first time point analyzed was P18-P21. At what age does one first see the *Fgfr3*-Ach phenotype, especially the increase in expansion of the resting zone chondrocytes and how much time since then does one see a size/bone phenotype. A temporal analysis that includes late embryonic and neonatal time points is strongly suggested.

<Authors' response>

We have added the histology of the proximal tibia of *Fgfr3^{Ach}* mice from 15.5 days postcoitum (E15.5) to P28 in Supplementary Figures 1 and 2. We also performed immunohistochemistry for Col10, a marker for the boundary between the secondary ossification center and growth plate (Supplementary Figure 3). Formation of the secondary ossification center was confirmed by expression of COL10 at the periphery of the secondary ossification center in *Fgfr3^{Ach}* mice at P18.

The heights of resting, proliferative, and hypertrophic zones were not respectively different between *Fgfr3^{Ach}* mice and control mice at P18 (Supplementary Fig. 1). We added this description to the Results section (lines 142-144). The height of the growth plate cartilage was significantly reduced in *Fgfr3^{Ach}* mice compared to that in control mice at P21.

2. Is the expansion in resting zone chondrocytes seen also in other mouse models that have investigated *Fgfr3*-Ach or is it unique to this mouse created by this lab? If it is unique

then it would suggest that the genetic background of the mice is also contributing to this phenotype and not just the p.Gly374Arg mutation. In this mouse model, the description provided by the authors suggest that the genetic background is essentially C57BL/6. Is this different from other Ach models created by other labs. The authors do not discuss why this is not seen in other labs as it is a very distinguishing feature of their mouse.

<Authors' response>

We agree that the resting zone phenotypes were not described in the text in previous reports. Therefore, we looked into the Figures in previously reported papers.

In the study by Shazeeb et al, [_ \(https://doi.org/10.1038/s41598-017-18801-0\)](https://doi.org/10.1038/s41598-017-18801-0)[1], Figure 6G indicates the histology of 3-week-old proximal tibia growth plates. Proliferative (red bracket) and hypertrophic (yellow bracket) zones in Ach mouse compared to WT mouse, respectively, are indicated. As seen in Figure 6G, the length of the resting zone above the red brackets appears slightly longer in the Ach mouse than in the WT mouse. Ach model mouse used in this study is the transgenic model expressing mouse *Fgfr3* containing the achondroplasia mutation under the *Col2* promoter[2]. The phenotype of this transgenic mouse model is milder than that of human achondroplasia, probably because *Col2* promoter did not recapitulate the expression patterns of *Fgfr3* gene. This would account for the reason why the degree of expansion of the resting zone is undetectable or mild in this transgenic mouse model. In addition, the difference

between FVB/NJ background in the transgenic mouse model and C57Bl/6 background in our study may also account for the reason.

Figure 6G obtained from the study by Shazeeb et al. ([https://doi.org/10.1038/s41598-017-](https://doi.org/10.1038/s41598-017-18801-0)

[FIGURE REDACTED]

18801-0),

Wang et al. (<https://doi.org/10.1073/pnas.96.8.4455>) reported conditional knock-in mice where G374R (the ortholog of human G380R) was introduced into Fgfr3 locus [3]. The mice in Wang et al.'s study are almost identical to ours except for that they used Cre/loxP system while we used Flp/frt system to make knock-in mice conditional. In Wang et al.'s study, Figures 5 c,d indicate the histology of 30-day-old proximal tibia growth plates. Although Wang et al. do not mention in the text, the resting chondrocyte zone appears markedly expanded in the knock-in mice (Figure 5d, red bracket which we added to the original figure) compared to that of the wild-type mouse (Figure 5c, red bracket which we added to the original figure). The histology of Wang et al.'s 30-day-old knock-in mice is similar to that of our 21-day-old knock-in mice shown in Figure 1h in our manuscript. The difference between the 129/SV background in Wangs' knock-in mice and the C57Bl/6 background in our mice may account for the difference in ages when the mice show typical phenotype.

Although the authors did not mention it in the text in previous studies, their figures indicate that the previously reported Ach model mice also showed expansion of the

resting chondrocyte zone similar to the histology that our mice showed. We have revised the description in the Results section (lines 149-151) and added this discussion to the Discussion section (lines 385-388).

[FIGURE REDACTED]

Figures 5 c,d obtained from the study by Wang et al. (<https://doi.org/10.1073/pnas.96.8.4455>)

Histological analysis of the growth plates from the tibia of neo-FGFR3 heterozygotes (FGFR3^{G374Rneo/+}) (d) and wild-type mice (c). We added resting zone (RZ) red brackets.

Our manuscript, Figure 1h, the second row.

Histological images of growth plate cartilage in the proximal tibia of *Fgfr3^{Ach}* mice (right) and control mice (left) at P21. Dotted lines indicate the resting, proliferative, and hypertrophic zones. SOC, secondary ossification center; RZ, resting zone; PZ, proliferative zone; HZ, hypertrophic zone; POC, primary ossification center.

3. The effect on the resting zone is potentially very interesting; however, the resting zone is poorly defined. Additional markers for resting zone chondrocytes should be used to determine if they are really looking at resting zone expansion and where they are drawing

their boundaries. In situ hybridization or immunostaining for PthrP or Gas1 (see Hallet et al., eLife 2021 10.7554/eLife.64513) in control and Ach mice should be used to confirm expansion of the resting zone. How does the expression of PthrP and Gas1 compare to that of Spondin1 in Fgfr3-Ach mice?

<Authors' response>

We attempted to perform immunohistochemical staining for PTHrP and GAS1 but had difficulty getting signals. Instead, we analyzed the expressions of *Clu*, *ApoE*, and *Sfrp5* using scRNA-seq data. Cluster #3 preferentially expressed these markers for resting zone chondrocytes. We also performed immunohistochemical analysis for SFRP5. SFRP5 was specifically expressed in the expanded resting zone in *Fgfr3^{Ach}* mice and in the resting zone of control mice. These results collectively confirm the hypothesis that cluster #3 corresponds to cells in the expanded resting zone of chondrocytes in *Fgfr3^{Ach}* mice. We have added these results in the Results section (lines 274-279) and in Supplementary Figure 10.

Regarding *PthrP* and *Gas1* mRNA expression, scRNA-seq analysis detected preferential expression of *Pthlh* (Supplementary Fig 12 a,b) and *Gas1* (Figure below) in clusters #3 and #2.

4. Resting zone chondrocytes may have some stem cell properties, but they are chondrocytes and not stem cells and the authors need to keep this mind when writing about this zone and not equate the two. The cells in this zone are chondrocyte progenitors at best but not stem cells. Also, a stem cell usually is not defined by the presence of one marker but a combination of markers.

<Authors' response>

We agree that resting chondrocytes are not stem cells. They have some stem cell properties such as self-renew activities to limited extent. We have substituted the term “stem cell-like properties” for “stem cell properties” throughout the manuscript.

5. From the ScRNAseq clusters, they essentially ignore Cluster 2 even though, like Cluster 3, it is also enriched in mutant cells. From Sup Fig 3b one can see that Cluster 2

shows the biggest change in numbers in all 3 mice that were used for this analysis and not cluster 3. Some analysis and discussion of cluster 2 should be provided.

<Authors' response>

Thank you for this important point. In addition to cluster #3, cluster #2 was enriched with *Fgfr3^{Ach}* mouse cells (Fig. 4a; Supplementary Fig. 8). To characterize cluster #2, we performed immunohistochemical analysis for markers for cluster #2 and RNA velocity assay. DEGs between cluster #2 and the remaining clusters included *Colla2* and *Omd* (Supplementary Fig. 11a). The *Omd* gene encodes OSTOMODULIN (also known as OSTEOADHERIN). The feature plot function confirmed that DEGs were preferentially expressed in cluster #2 (Supplementary Fig. 11b). Immunohistochemical analysis using an anti-COL1 or anti-OSTEOMODULIN antibodies indicated that they were specifically expressed in the boundary between the resting zone and secondary ossification center in both the control and *Fgfr3^{Ach}* mice (Supplementary Fig. 11c). Cells in cluster #2 expressed *Coll10a1* (Supplementary Figs. 7c, 11b), and cells at the boundary between the resting zone and secondary ossification center were hypertrophic and expressed COL10 (Supplementary Fig. 3). These results suggest that cluster #2 corresponds to cells in the boundary between resting zone and the secondary ossification center. We selected clusters #2 and #3 cells and subjected them to trajectory inference and RNA velocity analysis. The result indicated that cluster #3 become cluster #2 (Supplementary Fig 9d). These results collectively suggest that primary lesion resides in cluster #3 rather than #2. We have added this information to the Results section (lines 281-296) and Supplementary Fig. 11.

6. The authors need to follow the increase of *Fgfr3* activation or its decrease in the rescued mice by looking at additional downstream targets of *Fgfr3* such as *Stat1* or *Snail*.

<Authors' response>

We performed additional expression analysis for FGFR3, *Stat1*, and *Stat5*. scRNA-seq analysis indicated that *Fgfr3* was highly expressed in cluster #3 cells in *Fgfr3^{Ach}* mice compared to those in control mice (Supplementary Fig. 12 a,b). Immunohistochemistry confirmed the expression of FGFR3 in the expanded resting zone chondrocytes in *Fgfr3^{Ach}* mice (Fig. 5a). FRS2, a direct substrate of fibroblast growth factor receptors (FGFR), was phosphorylated in chondrocytes within the expanded resting zone of *Fgfr3^{Ach}* mice (Fig. 5a). Activated FGFR3 signaling increases expression of STAT1 and STAT5 in chondrocytes[4]. Our scRNA-seq analysis revealed that *Stat1* was expressed

slightly and *Stat5a* and *Stat5b* highly expressed in cluster #3 in *Fgfr3^{Ach}* mice (Supplementary Fig. 13). Immunohistochemistry confirmed preferential expression of STAT5 in the expanded resting zone chondrocytes in *Fgfr3^{Ach}* mice (Supplementary Fig. 14). These results indicate that FGFR3 is highly expressed and FGFR3 signal is activated in the expanded resting zone chondrocytes in *Fgfr3^{Ach}* mice. We have added this information to the Results section (lines 300-313), Figure 5a, and Supplementary Figs. 12, 13, 14.

We found that the administration of 666-15 reduced expression of STAT5 in resting zone chondrocytes in *Fgfr3^{Ach}* mice. We have added this information to the Results section (lines 365-367) and in Supplementary Fig. 6f.

7. With few exceptions, the IHC experiments are not very visually convincing. The choice of colorimetric IHC shows very weak expression and it is very hard to see the positive signals because the contrast of the positive signals with the secondary stain is not very high. In several cases it is very difficult to see a signal above background (CBP, p-FRS2). Spondin1 IHC in Fig. 6 is not convincing.

<Authors' response>

We performed immunohistochemical analysis to obtain improved images. We substituted Spondin1 in Figures 4d and 6f, p-FRS2, p-CREB, and CBP in Figure 5a with improved images.

8. The model says that resting zone chondrocytes are unable to proliferate. The experiments with EDU probably influenced this conclusion. But this is not consistent with expansion of the resting zone. Some discussion of this point and maybe refinement of the model would help. Where are these cells coming from? Can they be identified in a lineage trace? This is the reason why a temporal experiment showing how the morphology and expression of resting zone markers changes with time is important to understand how resting zone expansion is taking place.

<Authors' response>

Although we performed histological analysis at different time points (Supplementary Figs. 1 and 2) and lineage tracing experiments, we were unable to determine where the increased number of resting zone chondrocytes come from *Fgfr3^{Ach}* mice. RNA velocity assay suggests that cluster #2 cell are progenies but not ancestors of cluster #3 cells (Supplementary Fig. 11d). Therefore, we have been considering that resting zone

chondrocytes remain within the resting zone rather than exit into proliferative zone in *Fgfr3^{Ach}* mice, resulting in the expansion of the resting zone. The model we propose in Fig. 3c has not been proved and is merely a proposal. We have revised the sentence to state that the model shown in Fig. 3c is just a proposal (lines 195-196).

9. The DISCUSSION Section is very weak. It is necessary to discuss these results in the context of other models for *Fgfr3- Ach* and how the current model is similar and/or different.

<Authors' response>

As described in our response to the comment #2 above, we have stated that the expansion of the resting zone could be recognized in the previously reported model mice despite not being discussed in those studies (lines 385-388). We have also stated the difference between previously reported vosoritide that target MAPK pathway in the proliferative and hypertrophic zone and our 666-15 that target the CREB pathway in resting zone (lines 405-410).

10. Figure 6a. Control WT Data is missing. Need to know if the rescue is partial or complete. Figure 6b suggests a partial rescue.

<Authors' response>

We have added body weights of control mice to Figure 6a. The rescue was partial. We have added this information by adding the word “partially” in the Results section (lines 358, 360, 373, and 1003).

11. Control data is missing in Fig 6 for administration of 666-15 to WT mice. If 666-15 is administered to WT mice (as shown for *Fgfr3-Ach* in Fig 6F), do control WT mice show more expression for CD73?

<Authors' response>

We administered control mice with the CREB inhibitor 666-15 at a dose of 10 mg/kg from P7 to P27 and sacrificed them at P28, under the same protocol we employed for *Fgfr3^{Ach}* mice. Administration of 666-15 significantly changed neither weight, femur length, nor expression of CD73 in resting zone in control mice (Supplementary Fig. 16), suggesting that the 666-15 at a dose of 10 mg/kg is not effective in a physiological condition or that CREB activity is dispensable in chondrocytes in a physiological

condition. We have added this information to the Results section (lines 370-373) and the Methods section (line 683) and added data to Supplementary Figure 16.

12. Sentence 816: Figure 6 legend should say “666-15 partially rescued the Phenotype” as the bones showed elongation but did not reach WT levels in Fig 6b.

<Authors’ response>

We have corrected the legend for Figure 6 as follows: “..... 666-15 partially rescued the phenotype of *Fgfr3^{Ach}* mice” (lines 1003-1004).

13. Fig 6e. The immunostaining for Spondin1 is essentially not visible to the naked eye. The authors need a better figure. I suggest that they try immunofluorescence and confocal imaging for all their immunohistochemistry analysis.

<Authors’ response>

Thank you for your helpful suggestion. We performed immunohistochemistry for spondin1 using immunofluorescence and obtained improved images. We substituted the images in Figure 6f.

14. It would have been interesting to see if any cells in the expanded resting zone in *Fgfr3^{Ach}* mice are dying by apoptosis. Did the authors perform any TUNEL analysis of the growth plate?

<Authors’ response>

We additionally performed TUNEL assay and did not find increased staining in resting zone chondrocytes in *Fgfr3^{Ach}* mice compared to that in control mice. This result does not suggest that depletion of *Fgfr3^{Ach}* resting zone chondrocytes at P28 was due to apoptosis. We added data on TUNEL assay to the Method section and Supplementary Figure 5, and added these results and speculation to the Results section (lines 211-214; 217-221).

Minor Issues:

1. Sentence 48-49: “within the growth plate cartilage”. This reads like a compound Statement. Also, statement is scientifically wrong.

<Authors’ response>

We agree that it is a compound statement and scientifically wrong because endochondral bone formation includes the primary ossification center. We deleted the phrase “within the growth plate cartilage” and rewrote the sentences as follows (line 48-50): “Bone growth occurs through a process called endochondral bone formation, in which chondrocytes undergo sequential differentiation into three spatially arranged layers in growth plate cartilage:”.

2. Sentence 53, 151, 194: As pointed out above, resting zone chondrocytes are not stem cells.

<Authors' response>

As described in our response to the comment #4, we substituted the term “stem cell-like properties” for “stem cell properties” throughout the manuscript.

3. Sentence 163: “by mating *Fgfr3*^{Ach} mice: The authors did not mate Ach mice as these mice die by 6 weeks. They introduced the *Col11a3-CreERT* in the parents' mating profile. Sentence needs to be corrected.

<Authors' response>

We corrected the sentence as follows (lines 182-185): We crossed mice to generate *Fgfr3*^{Ach} mice bearing chondrocyte-specific *Col11a2-CreERT* transgene and multicolor *R26R-Confetti* reporter gene”.

4. 176: The authors need to add here that the schema is shown in Fig 3C.

<Authors' response>

We cited a reference in the first sentence of the paragraph for Figure 3c (lines 195-196): “Based on the results from EdU labeling and lineage tracing experiments, we propose following the model whose scheme is shown in Fig. 3c.”.

5. 188: how do the authors know that it is a partial closure? Is this conjecture or did they investigate this?

<Authors' response>

We did not investigate whether the growth plate in *Fgfr3^{Ach}* mice at P28 is closing or not. We deleted the phrase “, ultimately leading to partial premature closure of the growth plate”.

6. 198: not “resting” but “resting zone chondrocyte abnormalities”.

<Authors’ response>

We corrected the sentence by inserting the word “zone” (line 225).

7. 208: perhaps more appropriate to say “hematopoietic markers” as opposed to blood cell markers.

<Authors’ response>

We corrected the phrase to “hematopoietic markers” (line 235).

8. 219: not just Cluster 0 but cluster 1 also has strong expression of markers Sp7 and Pth1r as is evident from Figure. So cluster 1 is unlikely to be proliferative chondrocytes.

<Authors’ response>

Some cells in cluster #1 expressed Sp7 and Pth1r while other cells in cluster #1 did not. These results suggest that that cluster #1 is likely associated with prehypertrophic zone chondrocytes and proliferative zone chondrocytes. Accordingly, we corrected the description of cluster #1 in the Results section (lines 250-252).

9. 226: “enriched in” rather than “enriched with”.

<Authors’ response>

We corrected “enriched with” to “enriched in” throughout the manuscript (lines 243, 246, 254-255, 257, and 1109).

10. 255: hypertrophic zone: not shown in Figure

<Authors’ response>

We drew lines that demarcate resting, proliferative, and hypertrophic zones to indicate each zone in Figure 5a.

11. 272: Typo: not “maker” but “marker”

<Authors’ response>

We corrected “maker” to “marker” (line 344).

12. 278: not FGFRR.

<Authors’ response>

We corrected “FGFRR” to “FGFR3” (line 350).

13. 470: The cells were Lysed, plated....?? Why were the cells lysed. If cells were lysed, what did the authors grow/plate?

<Authors’ response>

We corrected “lysed” to “thawed” (line 614).

14. 727: 5 mm in (a)? or (b)?

<Authors’ response>

We corrected “(a)” to “(b)” (line 894).

15. 731: Stem cell chondrocytes. See above. Stem cells are not chondrocytes. This is a misnomer.

<Authors’ response>

We deleted the phrase “stem cell” from the sentence.

16. 779: The authors need to write what color is Spondin1 after the IHC? Brown? Blue? What should we look for?

<Authors’ response>

We performed immunohistochemical analysis for Spondin1 using fluorescence to obtain improved images. We have presented the term "Spondin1" in red in Figures 4d and 6f.

17. Figure 5g: there is a qPCR analysis shown. But its description is missing in legends (Sentence 806-809)

<Authors' response>

We are afraid that this may have been misconstrued. Fig. 5f shows qRT-PCR analysis and Fig. 5g shows western blot analysis.

18. Supplementary Fig. 3. 875: How was the ratio calculated? What was the denominator?

<Authors' response>

We explained how the ratio was calculated in detail in the legend of Supplementary Figure 8c as follows (lines 1103-1106): For each sample (d19g.ach.4, d19g.cont.3, d22d.ach.2, d22d.cont.1, d22e.ach.5, and d22e.cont.6), the number of cells in each cluster was divided by the total number of cells in the sample. The resulting values are shown as a percentage.

References

1. Shazeeb, M.S., Cox, M.K., Gupta, A., Tang, W., Singh, K., Pryce, C.T., Fogle, R., Mu, Y., Weber, W.D., Bangari, D.S., Ying, X., and Sabbagh, Y., Skeletal Characterization of the Fgfr3 Mouse Model of Achondroplasia Using Micro-CT and MRI Volumetric Imaging. *Sci Rep*, 8(1): p. 469, 2018.
2. Naski, M.C., Colvin, J.S., Coffin, J.D., and Ornitz, D.M., Repression of hedgehog signaling and BMP4 expression in growth plate cartilage by fibroblast growth factor receptor 3. *Development*, 125(24): p. 4977-4988, 1998.
3. Wang, Y., Spatz, M.K., Kannan, K., Hayk, H., Avivi, A., Gorivodsky, M., Pines, M., Yayon, A., Lonai, P., and Givol, D., A mouse model for achondroplasia produced by targeting fibroblast growth factor receptor 3. *Proc Natl Acad Sci U S A*, 96(8): p. 4455-4460, 1999.
4. Legeai-Mallet, L., Benoist-Lasselin, C., Munnich, A., and Bonaventure, J., Overexpression of FGFR3, Stat1, Stat5 and p21Cip1 correlates with phenotypic severity and defective chondrocyte differentiation in FGFR3-related chondrodysplasias. *Bone*, 34(1): p. 26-36, 2004.

Sincerely yours,
Noriyuki Tsumaki

Detailed Point-by-Point Response to Reviewers' Comments

Dear reviewers,

We would like to thank all the reviewers for their time and helpful comments. Following these comments, we have made changes to the manuscript (presented in red font). Below are our point-by-point responses (in black font) to the reviewers' comments (in blue font).

Reviewer #1 (Remarks to the Author): The authors have addressed all of my concerns in a clear, thorough, and detailed manner. In my view, the manuscript has been substantially improved. It now reads more smoothly; the data are presented more comprehensively; the figures are of higher quality; and the additional analyses enable more robust conclusions. The clarified figure presentation further enhances readability. My congratulations to Tsumaki-sensei and his team for this rigorous revision, the exciting narrative, and the important discoveries it contains. I hope the authors will continue this line of research and explore combinatorial therapeutic approaches.

I now find the evidence very convincing that achondroplasia is associated with impaired behavior of progenitor cells in the resting zone. This is a highly novel and important finding, shedding new light on the mechanisms by which FGFR3 activation leads to growth retardation. The mechanisms uncovered here may ultimately support the development of new therapeutic approaches for children with achondroplasia.

<Authors' response>

We sincerely appreciate the reviewer's helpful and insightful comments. We believe that our manuscript has been improved.

Reviewer #3 (Remarks to the Author): Overall, the manuscript is significantly improved and the revised IHC data is superior in quality and allows for improved confidence in the observation that it is the resting zone in Fgfr3Ach mouse growth plate that is enlarged, a major conclusion from this paper. This observation was also helped by the ScRNAseq data set that identified Spondin1 as a unique marker of the resting phase. The improved IHC allowed for better visibility of the enlarged resting phase and also increased confidence in this conclusion. Likewise, the new Stat5ab IHC shown in this manuscript also supports this observation.

<Authors' response>

We are truly grateful for the reviewer's valuable and perceptive feedback. We believe that our manuscript has significantly improved as a result.

The other major conclusion is the identification of the CREB pathway as being activated in the resting zone and its contribution to preventing chondrocytes from differentiation into proliferative chondrocytes. However, the authors are also modeling that the CREB pathway is inhibiting the self-renewal properties of the resting zone chondrocytes. But if the CREB pathway is inhibiting self-renewal then EdU incorporation into the resting zone as seen in Figure 2 would not have increased.

The long term EdU labeling in Figure 2 Control also does not make sense. Where do the EdU labeled proliferating chondrocytes go. At least some should be seen in the hypertrophic zone or in trabecular osteoblasts that are derived from hypertrophic chondrocytes.

From analysis of ScRNAseq data, the authors also conclude that cluster #3 (resting zone cluster) gives rise to cluster #2, which means that the cluster #2 zone is not contributing to the resting zone. The resting zone is enlarged, and these cells must come from somewhere. The increased EdU in this zone suggests self-renewal and failure to differentiate into proliferating chondrocytes, which is at odds with the CREB model in figure 7 of decreased self-renewal. It seems that the model in Figure 7 is not consistent with the model in Figure 3c, suggesting that the Figure 7 model is not correct. The only conclusion that seems supported by the data is that the CREB Pathway is preventing the resting zone chondrocytes to differentiate into proliferative chondrocytes which results in enlarging the resting zone.

<Authors' response>

Thank you for your valuable comments, which helped us clarify the interpretation of our data. In the long-term EdU labelling assay, only cells that divided when they incorporated EdU and subsequently divided very slowly (or entered dormancy) retain EdU label [1]. The amount of label retained in cells that actively undergo cell division (proliferation), such as proliferative chondrocytes and osteoblasts, gradually diminishes as they divide repeatedly. In our long-term EdU labelling assay, the number of label-retaining cells in *Fgfr3^{Ach}* mice was more than that in control mice, suggesting that the resting zone chondrocytes in *Fgfr3^{Ach}* mice divide more slowly than those in control mice from P7 to P21. We agree that the slow cell division of resting zone chondrocytes contributes to the

reduction in the expanded resting zone in *Fgfr3^{Ach}* mice. Prevention of the differentiation of resting zone chondrocytes into proliferative chondrocytes contributes to the expansion of the resting zone in *Fgfr3^{Ach}* mice. To make this point clearer, we corrected description in the results section (lines 211-213) as follows: “although resting zone chondrocytes underwent cell division slowly, continuous accumulation of progeny within the resting zone resulted in the expansion of the resting zone in *Fgfr3^{Ach}* mice by P21 (Fig. 3c, *right*)”. We also rewrote sentences in the Results section (lines 197-201; 208-209; 213-214). In addition, we deleted the word “dormant” or replaced the word “dormant” with “slowly dividing” throughout the manuscript, in order to interpret the results of the EdU labeling assay precisely.

Accordingly, we corrected Fig. 3c, right panel (*Fgfr3^{Ach}* mice). The word “Stop” was replaced with “Accumulation of cells in the resting zone”. More cells were added to the resting zone, suggesting the remaining and accumulation of progenies of resting zone chondrocytes in the resting zone. We also corrected Fig. 7 in which we deleted the phrase “Self-renew & Differentiation”, to make consistency between Fig. 3c and 7.

In the long-term EdU labelling assay, proliferative zone chondrocytes lose EdU label quickly after fast cell divisions. Only a few resting zone chondrocytes that proliferate very slowly retain EdU label. We consider that our results from the long-term EdU labelling assay in control mice in Figure 2b right panels are consistent with previously published findings (Fig. 5e, right panel) by Hallett et al. [2].

We modified discussion on the reduction of expanded resting zone in *Fgfr3^{Ach}* mice from P21 to P28. We speculate that slow cell division of resting zone chondrocytes could be one of the reasons why the expanded resting zone was reduced in *Fgfr3^{Ach}* mice (lines 221-227).

Even though the authors claim in their rebuttal that loss of self-renewal is only a model in Fig 3c, the abstract (line 34) claims this as a conclusion. This needs to be modified.

<Authors’ response>

Thank you for your valuable comments. We deleted the phrase “self-renew” from the Abstract and rewrote it as follows: “EdU labeling and lineage tracing analyses indicated that disruption of turnover and impairment of stem cell-like behavior of resting zone chondrocytes resulted in accumulation of cells in the resting zone.”

The authors have also included embryonic data and data of neonatal mice that show that the enlargement of the resting zone does not begin until after P18. A significant

observation as the transition from normal resting zone to enlarged resting zone is happening much later in neonatal mice. But there is no explanation for this phenomenon or even discussion of this observation in the Discussion section. Other mouse models for achondroplasia show changes in growth plate histology at earlier ages. What happens to the CREB pathway in neonatal mice before P18 when expansion of resting zone is not seen? Is the CREB pathway normal. This should be tested by IHC as the authors have done in Fig. 5a. Negative data from this analysis would further validate the involvement of the CREB pathway in this observation.

<Authors' response>

Thank you for your valuable and important comments. Our *Fgfr3^{Ach}* mice were smaller than the control mice from around P7 (Figs. 1c and 6a). We carefully measured heights of each zone, with referencing immunohistochemical data for markers such as Spondin1 and Stat5a/b at P8 and P18. The heights of proliferative zone was slightly reduced in *Fgfr3^{Ach}* mice compared with those of control mice at P8 (Supplementary Fig. 1b). After the secondary ossification center was formed, the height of growth plate cartilage was not significantly different between *Fgfr3^{Ach}* mice and control mice at P18. However, close examination revealed that the height of resting zone was slightly increased, and the height of proliferative zone was slightly decreased in *Fgfr3^{Ach}* mice compared with those of control mice at P18 (Supplementary Fig. 1a,c). The expansion of the resting zone became remarkable at P21. We added data of P8 and corrected data of P18 in the Results section (lines 142-148; 156-157) and Supplementary Fig. 1b,c.

We also analyzed CREB activity in growth cartilage at P8 and P18 by immunohistochemistry. Expression of SPONDIN1, phospho-CREB, and STAT5a/b were not so much different between *Fgfr3^{Ach}* mice and control mice at P8 (Supplementary Fig. 17b). The number of resting zone chondrocytes expressing SPONDIN1, phospho-CREB, and STAT5a/b was slightly increased in *Fgfr3^{Ach}* mice compared to those in control mice at P18 (Supplementary Fig. 17b). Based on these results, we speculated that increased CREB activity and impaired behavior of resting zone chondrocytes might begin at earlier stages and become prominent at P21. The reason why CREB is activated after the formation of the resting zone (after formation of the secondary ossification center) remains to be investigated. We added these data and discussion in the Results section (lines 387-396) and Supplementary Fig. 17.

Minor:

1. Fig. 1d, should be Survival not Probability of survival
2. Fig 2 legend:
Line 919. The authors use Click-iT chemistry to identify EdU positive cells. But writing “immunostained with anti-EdU antibody” here is wrong.
3. Fig. 3b show where insets come from. 3d, define SOFG
4. Line 270 and 1148 and also elsewhere: “violin plots” and not “violine”
5. Line 284 Omd is Osteomodulin, not Ostomodulin
6. Legend missing for Supplementary Fig. 9c

<Authors' response>

Thank you for important points. We corrected our errors.

We moved Supplementary Fig. 9c to Supplementary Fig. 15c. We added legend of Supplementary Fig. 15c (lines 1186-1187), and we mentioned Supplementary Fig. 15c in the Results section (lines 337-338).

In addition, We have corrected the images in Supplementary Fig. 3, because their images and magnifications were incorrect.

References

1. Kobayashi, S., Takebe, T., Zheng, Y.W., Mizuno, M., Yabuki, Y., Maegawa, J., and Taniguchi, H., Presence of cartilage stem/progenitor cells in adult mice auricular perichondrium. *PLoS One*, 6(10): p. e26393, 2011.
2. Hallett, S.A., Matsushita, Y., Ono, W., Sakagami, N., Mizuhashi, K., Tokavanich, N., Nagata, M., Zhou, A., Hirai, T., Kronenberg, H.M., and Ono, N., Chondrocytes in the resting zone of the growth plate are maintained in a Wnt-inhibitory environment. *Elife*, 10, 2021.

Sincerely yours,
Noriyuki Tsumaki